

# Shifting water scarcities: Irrigation alleviates agricultural green water deficits while increasing blue water scarcity

Heindriken Dahlmann[1,2,3], Lauren S. Andersen[3], Sibyll Schaphoff[3], Fabian Stenzel[3,4], Johanna Braun[3], Christoph Müller[3] and Dieter Gerten[1,2,3]

[1]Integrative Research Institute on Transformations of Human-Environment Systems, Humboldt-Universität zu Berlin, Berlin, Germany
[2]Department of Geography, Humboldt-Universität zu Berlin, Berlin, Germany
[3]Potsdam Institute for Climate Impact Research (PIK), Member of the Leibniz Association, P.O. Box 60 12 03, D-14412 Potsdam, Germany
[4]Stockholm Resilience Centre, Stockholm University, Stockholm, Sweden

*Correspondence to*: Heindriken Dahlmann (heindriken.dahlmann@hu-berlin.de)

**Abstract.** Agricultural areas often experience green water scarcity – i.e. soil moisture limitation on crop growth – due to e.g. unfavourable soil texture, high potential evapotranspiration rates, poor or inefficient crop management, and fluctuations in meteorological conditions. Driven by the growing effects of climate change and the rising water and food demands of an increasing world population, agricultural green water scarcity is becoming an increasingly important phenomenon. In this global modelling study, a plant-physiology based indicator of green water stress is applied, that quantifies the ratio between soil moisture limitation and atmospheric water demand on agricultural areas. Results show that currently (2015–2019 average) 37% of the global agricultural area is green water stressed, where this ratio is >0.2. Hotspots are characterized by a high seasonal variability in stress conditions, and are mainly located in India and Pakistan, northern Sub-Saharan Africa, North Africa and southwestern Asia. Using an analogous blue water stress indicator – which relates human water use for households, industry and agriculture to available blue water resources in rivers, reservoirs and aquifers – current irrigation is shown to alleviate green water stress on 13% of the total agricultural area (207 Mha) but simultaneously increases the share of areas experiencing blue water stress by 12% (199 Mha). Moreover, on average 585 km$^3$ yr$^{-1}$ of water used for irrigation (22% of the total water use) is found to stem from surface water resources at the expense of rivers' environmental flow requirements. This shift in water stress types highlights the importance of jointly considering the interconnected green and blue water resources and stresses in pathways towards sustainable water use in agriculture.

## 1 Introduction

Green water resources available to agriculture – the plant available rainwater held in soils which sustains the growth of crops and pastures – account for 85-90% of the water consumed by agriculture and are therefore of immense importance for securing global food production (Rost et al., 2008; Rockström et al., 2009; Hoekstra and Mekonnen, 2012). Due to climate change and



the intensification of agricultural practices in response to the higher food demand of the growing world population, green water scarcity increases in many agricultural areas (He and Rosa, 2023; W. Liu et al., 2022). In the future, even if the 1.5°C climate mitigation target would be achieved, two thirds of the global rainfed cropland could be affected by green water scarcity, posing

a considerable threat to agricultural productivity and, consequently, global food security (He and Rosa, 2023).

Green water is especially indispensable on rainfed cropland, where it presents the only water resource (X. Liu et al., 2022). On irrigated cropland, blue water resources have the potential to alleviate green water scarcity (GWS) and mitigate crop exposure to such stress (Rosa et al., 2019). Therefore, green and blue water resources and limitations are not only strongly interconnected via the hydrological cycle but also through human interference. However, the practical implementation of irrigation often faces

hindrances, primarily due to concurrent challenges such as blue water scarcity (BWS; Kummu et al., 2016; Mekonnen and Hoekstra, 2016) or a lack of irrigation infrastructure sometimes called economic water scarcity (IWMI, 2007; Rosa et al., 2020). Besides, irrigation systems are often inefficient, and a substantial portion of current global irrigation occurs in an unsustainable manner, impacting the maintenance of environmental flow requirements (EFRs) of aquatic ecosystems, depleting groundwater resources, and leading to severe water pollution (Falkenmark et al., 2013; Jägermeyr et al., 2017; Rosa

et al., 2019; Dalin et al., 2019).

Blue water resources have long been the focus of water scarcity analyses since they are at the center of the competition between sectoral human water uses and environmental water requirements (Kummu et al., 2016; Mekonnen et al., 2016; Veldkamp et al., 2017) as well as human water stress under climate-change and population growth (Heinke et al. 2019). Discussions about the central role, potential (increasing) limitation, and sustainable use of green water were long absent, as its availability was

often taken for granted. While there are some integrated scarcity assessments, incorporating blue and green water resources (Rockström et al., 2009; Gerten et al., 2011; Rosa et al., 2020; W. Liu et al., 2022; X. Liu et al., 2022, Liu et al., 2025), most of them simply overlay the different individual scarcities, not explicitly considering the interlinkages between them in a dynamic and process-based manner. For a consistent analysis, an approach is needed that a) accounts for the balance of plant-available soil moisture and atmospheric moisture deficit, that determines GWS; b) quantifies how the addition of blue water

through irrigation ameliorates GWS; c) traces how this irrigation may increase BWS; and d) investigates whether sufficient blue water resources are in principle available to sustainably alleviate GWS, e.g. not tapping into EFRs. The balance of plant-available soil moisture and atmospheric moisture deficit is particularly suitable for a GWS indicator, as it directly determines the stress level that plants are exposed to, limiting photosynthesis and growth. While Rosa et al. (2020) provide a quantitative, globally applicable approach to distinguish GWS and BWS, to our knowledge no study has quantified where and at what

magnitude GWS has led to a shift towards higher BWS.

To address these research gaps, this study aims at analysing current spatial patterns and interlinkages of GWS and BWS related to agriculture, employing the LPJmL dynamic global vegetation, crop and hydrological model (Schaphoff et al., 2018, von Bloh et al. 2018, Lutz et al. 2019, Wirth et al. 2024). Versions of this model have demonstrated capability to compute green-blue freshwater resources and limitations in coupling with natural and agricultural vegetation dynamics and in response to

changes in climate, atmospheric CO2 concentration, land cover/land use change, and crop and water management (Rost et al.,



2008; Jägermeyr et al., 2015, Stenzel et al., 2019). A physiological GWS indicator is employed here, computed separately for each of the world's major crop types, taking into account the balance of soil moisture and atmospheric water demand, at daily time steps over the period 2015–2019, and on a global 0.5° grid. Based on this indicator, first, current GWS hotspots are identified. Second, BWS is calculated at daily timesteps for the same period and mapped on a monthly basis using an indicator which relates human blue water use (for households, industry and irrigated agriculture) to available blue water. Third, it is traced where and to what degree irrigation ameliorates GWS but at the same time increases BWS. Finally, the extent to which local blue water resources would be sufficient to buffer GWS without ecologically unsustainable appropriation of EFRs is quantified.

## 2 Methods

### 2.1 The Dynamic Global Vegetation Model LPJmL

LPJmL (Lund-Potsdam-Jena managed Land) simulates the growth and productivity of natural and agricultural vegetation with the coupled water, energy, nitrogen, and carbon pools and fluxes (for detailed model descriptions see Bondeau et al., 2007; Schaphoff et al., 2018; von Bloh et al., 2018). It further captures the spatial and temporal variations of these processes in response to climatic conditions and human interventions such as crop management and irrigation (Jägermeyr et al., 2015, 2017; Lutz et al., 2019; Herzfeld et al., 2021; Porwollik et al., 2022; Minoli et al., 2019, 2022). Simulations are performed at a spatial resolution of 0.5° (further specifying fractions of each grid cell assigned to different crop, irrigation and pasture systems, the remainder is simulated as dynamic natural vegetation), at daily time steps. Natural vegetation is represented by nine plant functional types (PFTs) and agricultural vegetation by 12 crop functional types (CFTs) and grassland/pastures. In this study, CFTs are the focus, including: temperate cereals, rice, maize, tropical cereals, pulses, temperate roots, tropical roots, sunflower, soybean, groundnut, rapeseed, sugar cane, in addition to an "others" category, which aggregates all crops not parameterised specifically as CFTs (see Schaphoff et al. (2018)). CFTs are considered to be either rainfed or irrigated, prescribed by a land-use input dataset (see below).

Each CFT is simulated with its own soil bucket, so that the irrigation water requirement is crop-specific and the green water supply not influenced by the other plants. Daily net irrigation is determined for each CFT based on the soil water deficit, the CFT-specific water demand given by atmospheric moisture deficit, and the efficiency of the specific irrigation systems (Jägermeyr et al., 2015, Schaphoff et al., 2018). Irrigation is applied to the field if CFT-specific water stress occurs (water supply being lower than water demand) and soil moisture falls below a CFT-specific irrigation threshold. LPJmL assumes withdrawal of irrigation water from available blue water, e.g. rivers, lakes, reservoirs and renewable groundwater, including from neighboring upstream cells if the computed blue water availability of the cell where the demand occurs is not sufficient (accounting for local water diversion schemes and possible mismatches between the input datasets on river topology and irrigated areas).





## 2.2 Simulation protocol and model runs

In this study, LPJmL version 5.9.25 was run with forcing of historical climate and land use for the period 1901–2019, preceded by a 3,500-year spin-up period in order to bring the PFT distribution and carbon and nitrogen pools into a dynamic prehistoric
equilibrium and a subsequent land use spin-up featuring historical land-use patterns since 1500 (see S1). Daily climate forcing data were taken from the GSWP-W5E5 dataset (Lange et al., 2021, see Table S1). Land-use input obtained from the LandInG 1.0 toolbox (Ostberg et al., 2023) is used by the model to simulate each CFTs' growing period and seasonal phenology, as well as crop production and yield. Availability of green water (and potentially added blue irrigation water) is computed separately for each CFT, based on their specific soil water budget. While CFTs do not compete for green water, irrigation water is
allocated based on local blue water availability within each grid cell (river inflow excluding upstream water use, as well as reservoirs and lakes). Irrigation is assumed to occur to the extent it can be met by this daily, grid-cell specific blue water availability ("limited irrigation scenario" ILIM, not additionally including long-distance transfers or fossil groundwater; see Rost et al., 2008). To isolate effects of irrigation and thereby derive the climate-driven effect on GWS, a "no-irrigation scenario" is also analysed (INO, performed for 1990–2019) that considers all agricultural land to be rainfed. For better
comparability of irrigated CFTs in INO and ILIM, the growing seasons of the concerned CFTs in INO are adjusted so that they retain their actual growing season length (as in ILIM) and do not adapt to the rainfed growing season. Furthermore, a groundwater buffer is implemented to represent subsurface water storage, which releases water at a fixed rate of 0.01 per day relative to the current buffer volume, thereby simulating a simplified groundwater discharge process.

## 2.3 Green water stress

The GWS indicator (see Fig. 1) is defined as the unitless ratio of plant-available soil water supply and atmospheric demand, per CFT and grid cell on a monthly basis (derived from the average of daily values):

$$GWS = 1 - \frac{S}{D} \text{ (Eq. 1)}$$

The plant-available soil water supply is calculated as follows:

$$S = E_{max} \cdot w_r \text{ (Eq. 2)},$$

where $E_{max}$ is the maximum daily transpiration rate (8 mm d$^{-1}$), and $w_r$ is the relative soil moisture of the root zone, scaled between 0 (no green or blue water available) and 1 (fully saturated).

The atmospheric demand represents the daily "optimal" transpiration of a given CFT (unconstrained by soil moisture limitation) and is defined according to the following hyperbolic function:

$$D = \frac{(1 - w) \cdot E_{eq} \cdot a_m}{(1 + \frac{g_m}{g_c})} \text{ (Eq. 3)},$$

where $w$ is the amount of energy used to vaporize the intercepted water in the vegetation canopy (fraction of the day the canopy is wet); $E_{eq}$ is the equilibrium evapotranspiration; $a_m$ is the Priestley-Taylor coefficient (1.391); $g_m$ presents a scaling factor 3.26 and $g_c$ the potential canopy conductance that would occur if soil moisture was not limiting.




The GWS indicator ranges from 0 to 1, with 1 indicating complete water stress, and 0 representing an unstressed condition where soils can provide adequate water to meet D. If S > D, we set S = D to ensure the result remains within [0,1]. If S < D,
the plants' physiological activities decline, impacting their transpiration, biomass production, as well as carbon sequestration (see details in Gerten et al., 2005, 2007, who used a similar indicator for natural vegetation).

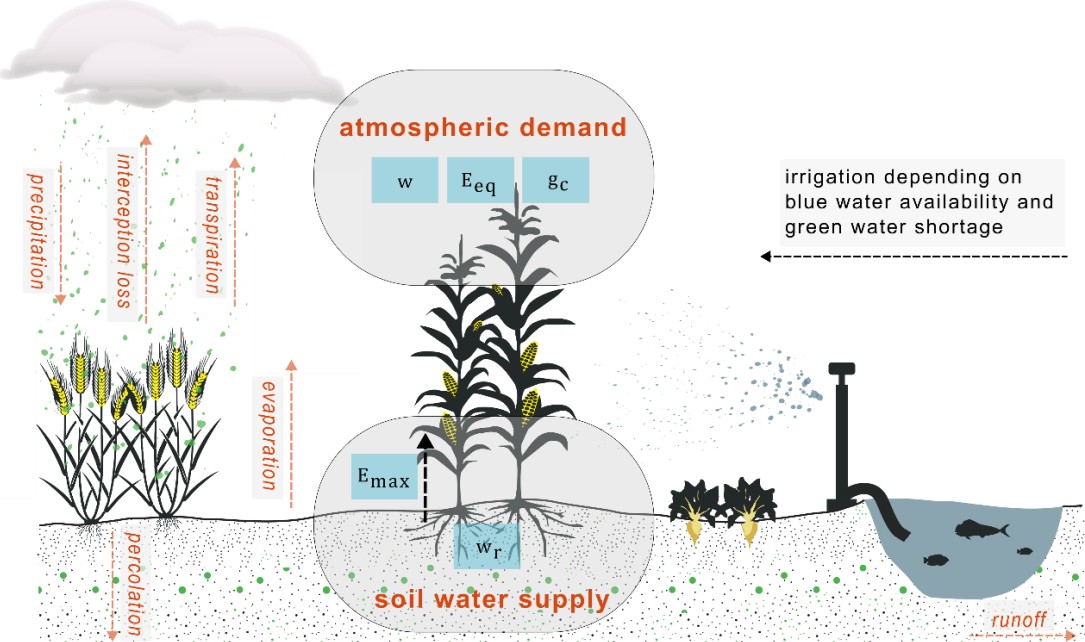

**Figure 1: Schematic overview of basic processes considered in the calculation of the green water scarcity indicator. See text/equations**
**for abbreviations.**

Monthly GWS is calculated for rainfed and irrigated CFTs based on the ratio of monthly sums of demand and supply, considering only days when the crop is actively growing. Days without any water demand are omitted from the aggregation. To calculate the average GWS for a grid cell, GWS was weighted by the annual CFT-specific area fraction and normalized by the
sum of all CFT fractions. Detailed maps of CFT-specific GWS can be found in the supplementary Material (see S14).
A CFT is considered green water scarce if GWS >0.2. This threshold was derived from literature, where it presents a commonly used approach, following the argument that yields decrease below this value and farmers apply irrigation where possible (He and Rosa, 2023). In this study, a CFT is considered highly green water scarce if GWS >0.4.

## 2.4 Blue water stress

Similar to the GWS, the BWS indicator presents a water use to availability ratio and is computed for each grid cell on a monthly basis comparing human water use (domestic, industrial and irrigation) to river discharge of the actual grid cell (Q) and water withdrawn from reservoirs ($WD_{res}$):





$$BWS = \frac{WU_{dom} + WU_{ind} + WU_{irr}}{Q + WD_{res}} \text{ (Eq. 4)}$$

Contrarily to the GWS indicator, the BWS indicator thus not solely considers water use for agriculture but from other sectors
as well (which however do not affect the difference in BWS between the ILIM and INO simulations as they are always
prioritized, using input from Flörke et al. 2013). A higher BWS indicates greater stress, reflecting the extent to which total
human water demand approaches or exceeds the renewable freshwater supply. Blue water stress is assumed to be moderate
(high) in cells where the yearly mean BWS is >0.2 (> 0.4) (Raskin et al., 1997; Vörösmarty et al., 2000).

## 2.5 Sustainable blue water use

EFRs are a method to quantify aquatic ecosystem water needs. Following Stenzel et al. (in preparation), we define blue water
use in agriculture as unsustainable water overuse, if it leads to transgressions of EFRs. We calculated EFRs for every grid cell
on a monthly level using the Variable Monthly Flow method, which accounts for the intraannual variability of river flows and
separates the flow regime into high, intermediate, and low-flow months (Pastor et al., 2014). Water overuse (O) was calculated
at the same resolution by subtracting human water use (domestic, industrial and irrigation), as well as EFRs from the available
discharge (Q):

$$O = Q - (WU_{dom} + WU_{ind} + WU_{irr}) - EFRs \text{ (Eq. 5)}$$

The resulting water overuse values were then aggregated across all agricultural grid cells to derive a global estimate.

## 3 Results

### 3.1 Global hotspots of green water stress

To exclude the ameliorating impact of irrigation, average GWS of all CFTs was assessed in the INO scenario that assumes all
agricultural areas to be rainfed for the time period 2015–2019 (Fig. 2a). Major GWS hotspots with values close to 1 – in the
absence of irrigation – are found to be located in southwestern Asia and North Africa, particularly in Iran, Afghanistan,
Pakistan, Egypt and on the Arabian Peninsula. Other, albeit less green water stressed regions with values >0.6, are located in
northern Sub-Saharan Africa, e.g. Sudan, Somalia and Niger, and southern Africa where local communities are highly
dependent on rainfed agriculture. Also, regions in Mongolia and Kazakhstan, eastern Australia, South America, Mexico and
the USA experience values of GWS >0.6. Regions with little GWS (annual average <0.2) even without irrigation include large
areas of Europe, parts of eastern China, rainforest regions, and parts of the US. In these regions, soil water supply is able to
meet atmospheric water demand in most months of a year. In the INO scenario, 50% of the global cropland (which equals 799
Mha) experiences green water stress >0.2 (see Table 1).





**Figure 2: LPJmL-simulated GWS in the absence of irrigation (INO scenario) across crop functional types for the time period 2015-2019 as a) yearly average and b) seasonal average. White areas indicate that no crops are growing in this period.**

The green water stress patterns show a high seasonal variability over the year due to changing weather conditions but also season-specific growing seasons (Fig. 2b). Europe and North America do experience less (or even no) GWS during the winter months since the water demand of the crops grown during that time is very low. During the summer months, however, especially southern regions in Europe and the western US are highly green water stressed (GWS >0.4). Large regions in Brazil change into GWS hotspots from June to November where pulses, rapeseed and sugarcane are especially green water stressed.





India, by contrast, does not experience high GWS from June to November, when most crops are grown. However, temperate
cereals, pulses, oil crops, and sugarcane, which are cultivated in the dry winter monsoon season, do experience GWS. Besides
the overall regional and seasonal features, the simulated GWS patterns also show differences in GWS between the individual
years (see Fig. S9 for seasonal maps of the years 2017, 2018 and 2019). For instance, the drought in large parts of Europe in
2018 is well captured, as are the generally very wet conditions in 2017 there (for a comparison see Fig. S10).

**Table 1: Calculations of the extent of water stressed areas and associated global water volumes.**

| | | **This study (2015-2019)** | **Other studies** |
|---|---|---|---|
| **All agricultural area** | | 1579 Mha | |
| Green water stress | | | |
| | 0.2 – 0.4 | INO: 512 Mha (32%) | |
| | | ILIM: 431 Mha (27%) | |
| | 0.4 - 1 | INO: 287 Mha (18%) | |
| | | ILIM: 161 Mha (10%) | |
| Blue water stress | | | No comparable global BWS studies with explicit focus on agricultural land. |
| | 0.2 – 0.4 | INO: 25 Mha (2%) | Comparable numbers: |
| | | ILIM: 125 Mha (8%) | - 1611 Mha (11%) of global land area experiences BWS > 0.2 (Stenzel et al., 2021) |
| | 0.4 - 1 | INO: 26 Mha (2%) | - 380 Mha (39%) of global croplands water scarce (blue and green water scarcity combined) [1981-2005] (X. Liu et al., 2022) |
| | | ILIM: 125 Mha (8%) | |
| **Rainfed area (ILIM)** | | 1302 Mha | |
| Green water stress | | | |
| | 0.2 – 0.4 | 426 Mha (33%) | 394 Mha (53%) [1996-2005, threshold: GWS > 0.2] (He and Rosa, 2023) |
| | 0.4 - 1 | 213 Mha (16%) | |
| **Blue water volume added to mitigate GWS (= irrigation withdrawal)** | | 2506 $km^3yr^{-1}$ | 2409 $km^3yr^{-1}$ [2000] (Jägermeyr et al., 2017); 2690 $km^3yr^{-1}$ (McDermid et al., 2024 [average of various studies and years]) |
| **Blue water added from non-sustainable surface water resources** | | 585 $km^3yr^{-1}$ | 611 $km^3yr^{-1}$ [1996-2005] (Rosa et al., 2020); 569 $km^3yr^{-1}$ [2015] (Rosa et al., 2019) |



## 3.2 Irrigation-induced shifts from green to blue water stress

While the above analysis assumed all agricultural areas to be rainfed (INO), we here compare the INO and the limited irrigation (ILIM) scenarios to firstly quantify the GWS-mitigating effect of irrigation and secondly the degree to which this irrigation increases BWS.

Table 1 shows that 50% (799 Mha) of all global cropland is green water stressed (GWS > 0.2) in the INO scenario, with 32% of the area (512 Mha) experiencing moderate GWS (0.2–0.4) and 18% (287 Mha) high GWS (0.4–1). With irrigation (ILIM scenario) this value decreases to 37% (592 Mha), where 27% are classified as moderately and 10% as highly stressed. This means that 13% of the global cropland area experiences GWS <0.2 due to the alleviating effect of irrigation.

Figure 3 illustrates the difference of GWS between the INO and ILIM scenario, averaged over the time period 2015-2019.
According to expectation, irrigation significantly ameliorates GWS in many regions, for example in Southwest Asia, India, Southern Africa and Peru. Figure 4a shows the simulated spatial patterns of combined GWS and BWS under the ILIM scenario. Figure 4b highlights that in the INO scenario, only 42% of the agricultural area in cells with irrigation experience no/low GWS (<0.2), but with irrigation (ILIM), this number increased to 72% (see Table S3). Correspondingly, the area in these cells facing moderate GWS drops from 36% without irrigation to 24% with irrigation, and for severe GWS from 23% to 8%, respectively.

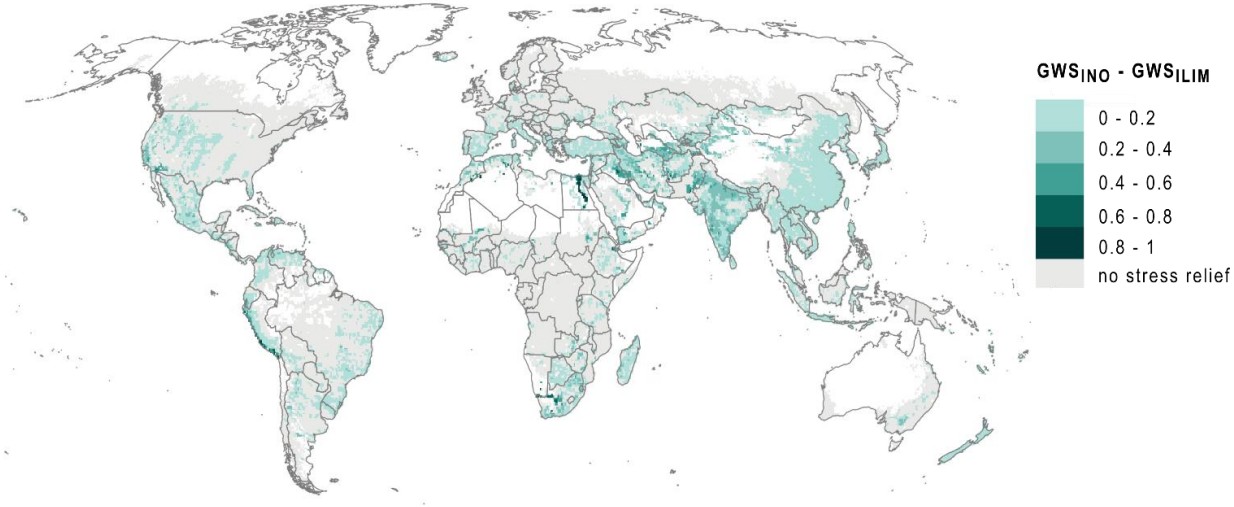


**Figure 3: Green water stress relief due to irrigation. LPJmL-simulated difference between values of GWS in the INO and ILIM simulations, averaged for the time period 2015–2019.**

Figure 4a furthermore shows that regions mainly experiencing BWS are in India, China, Southwest Asia, parts of the US,
southern Africa and the Mediterranean. In the ILIM run, we find that 8% (125 Mha) of all global agricultural areas (rainfed





and irrigated) experience moderate (BWS >0.2) and 8% (125 Mha) experience high (BWS >0.4) blue water stress (see Table 1). Comparing these results with the INO scenario (2% for both cases) confirms that the main reason for BWS is irrigation water use, as detailed in the following based on model results shown in Fig. 4a (see Fig. S11 for only BWS). A detailed look at agricultural areas in cells with irrigation (Fig. 4b) highlights again that the GWS-ameliorating effect leads to a shift from

GWS to BWS there. In the INO scenario, only 2% of the agricultural area in cells with irrigation experience moderate as well as high BWS, respectively (due to water extraction for industrial and domestic purposes), but with irrigation in the ILIM scenario, these shares rise to 17%, respectively (see Table S4). These shifts happen mainly in regions like India, Iran, Southern Africa but also Turkey or Spain.

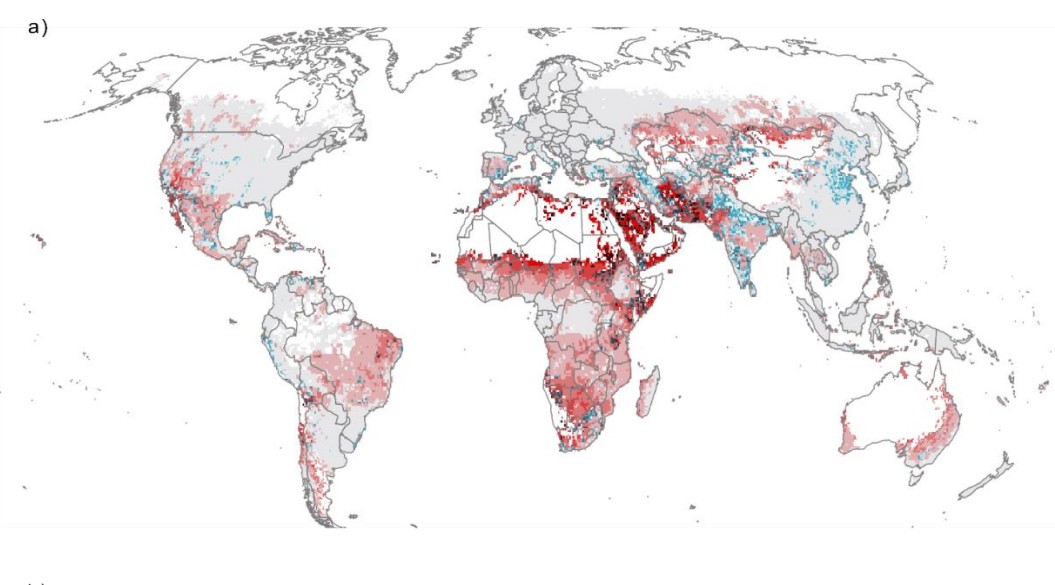

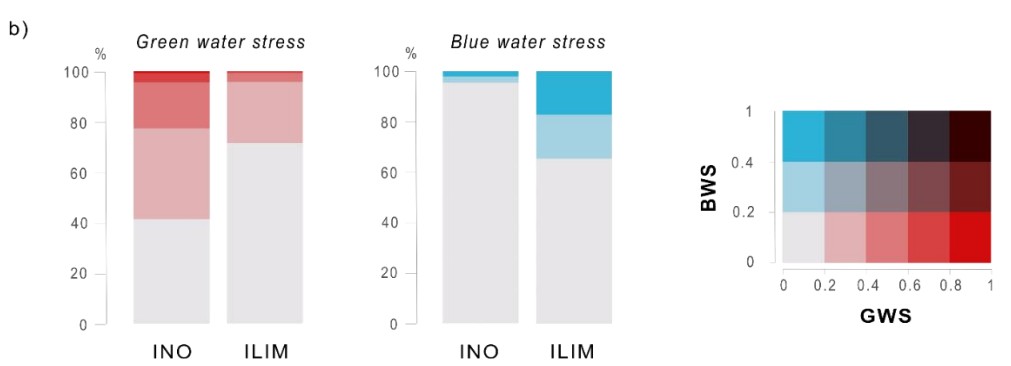


**Figure 4: Combined water stresses: a) simulated spatial pattern of combined GWS and BWS under ILIM, averaged over the period 2015-2019, and b) shifts in water stresses in agricultural areas in cells with irrigation from the INO to the ILIM scenario for GWS and BWS. The legend applies for both panel a) and b), where grey to blue shades along the y-axis indicate increasing BWS and grey to red shades along the x-axis indicate increasing GWS; for panel a, the diagonal line from light grey in the lower left to dark violet**
**in the upper right indicates increasing combined BWS and GWS.**



## 3.3 Sustainable use of blue water resources

In a final step, it is analysed whether sufficient blue water resources are available to sustainably alleviate GWS (i.e. decreasing the GWS indicator value to near 0) – which is a hypothetical scenario as in reality EFRs are often neglected. Figure 4a already shows that some areas that experience both GWS and BWS are located in southwestern Asia, Pakistan, Afghanistan, Somalia,

Brazil, Mexico and the US. The simultaneous stress of both green and blue water already indicates that GWS is often mitigated at the expense of blue water resources.

We find that during the study period, on average 585 $km^3yr^{-1}$ of blue water resources have been added non-sustainably, leading to the transgression of EFRs (see Fig. 5). This overuse accounts for 22% of the total water consumption (around 2700 $km^3yr^{-1}$), with particularly high volumes in India and China.

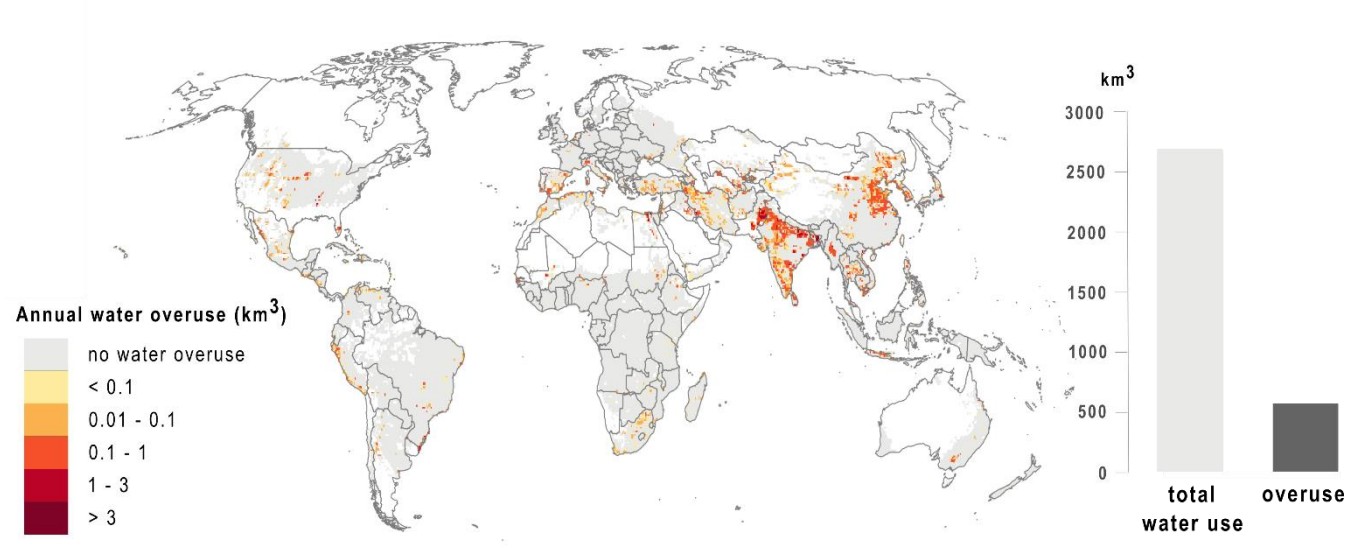


**Figure 5: Annual unsustainable water overuse on agricultural areas ($km^3$, average 2015-2019). The bars to the right indicate the globally aggregated volumes.**

## 4 Discussion

### 4.1 Green water stress indicator and hotspots

This study identifies current hotspots of GWS in agricultural regions on an annual and seasonal scale for the time period 2015–2019. The GWS indicator used focuses not solely on soil moisture, but on CFT-specific water demand and water supply and therefore may reflect low water stress even under low soil moisture conditions, as long as atmospheric evaporative demand is also low. In this analysis, GWS hotspots are defined as areas of plant-physiological stress, where regions are characterized as





green-water scarce when GWS >0.2, a threshold that was derived from literature (He and Rosa, 2023). While we added a second threshold (0.4–1) indicating severe GWS, this threshold-setting could be further improved by examining the relationship between crop yields and GWS more closely, enabling the definition of CFT-specific GWS thresholds.

In general, the GWS hotspots identified here are in line with spatial patterns of earlier studies, even though regional differences exist. Comparing the results of our GWS indicator to He and Rosa (2023), who computed GWS as the ratio of irrigation water requirements and crop water requirements for the baseline period 1996-2005 on rainfed areas (see Fig. S4), illustrates that their study shows more extreme GWS, as e.g. in the western USA or Europe. He and Rosa (2023) further identify 38% less rainfed area under green water stressed conditions compared to our findings (see Table 1). This discrepancy may be attributed to differences in the model and input data used, as well as the underlying total rainfed area (1302 Mha in our study and 555 Mha in He and Rosa, 2023). The relative values of both studies are however well comparable. In contrast to the frequently used actual to potential evapotranspiration (AET/PET) ratio, which rather represents an average situation across land systems, our GWS indicator shows that even though indicator values vary, patterns remain similar (see Fig. S5).

In LPJmL, multiple cycles of the same crop or different crops in one year are not implemented. The inclusion of multiple cropping would however provide a more accurate representation of agricultural practices in regions with multiple harvests per year and also be essential for properly assessing the benefits of irrigation. Previous work (e.g. Biemans et al., 2016) has shown that enabling multi-cropping in LPJmL led to better seasonal estimates of irrigation-water demand. We therefore likely underestimate irrigation water supply and thus the alleviation of GWS in this study. A second underestimation might arise from the ILIM setting of this study, where biases in LPJmL in reproducing the seasonal cycle of discharge might lead to artificial mismatches in water supply and irrigation water demand.

## 4.2 Interlinkages between green and blue water stress

Combining green and blue water stresses revealed that they are strongly interlinked through irrigation. While irrigation reduces GWS (below the threshold of 0.2) on 13% of the global agricultural area, it thereby leads to an increase in areas experiencing moderate BWS as well as high BWS by 6%, respectively. However, the shift in water stresses from green to blue is recognized to rely on the definition and simulation of the water stress indicators (see above for GWS), as well as irrigation assumptions in LPJmL.

The BWS indicator applied in this analysis has been applied in numerous studies (Vörösmarty et al., 2000; Smakthin et al., 2004; Kummu et al., 2016). In its original version, it is meant to reflect local availability by accounting for available water from river discharge only. It does not assume the sharing, transfer or trade of water. Irrigation in LPJmL, however, not only relies on local river discharge but also on water withdrawals from reservoirs and withdrawals from upstream neighboring cells. This issue was addressed by adjusting the BWS indicator, not only accounting for river discharge but also water withdrawals from reservoirs within the given grid cell. A detailed analysis of the differences in BWS calculations with and without accounting for reservoir withdrawals is provided in the supplementary material (Fig. S11 and S12). Comparing the results showed that including withdrawals from reservoirs reduced blue water stressed areas by 2% (see Table S4). For the calculation



of the BWS, it would be beneficial to consider not only water available in the given grid cell but also external water inflows. These could be water withdrawals from upstream cells or long-distance water transfers - such as those via irrigation canals - which often play a significant role in highly irrigated regions.

The results of this study are further influenced by regional factors of irrigation, such as different water use efficiencies, the precise location of irrigation systems or the type of irrigation applied. These variables have a strong influence on BWS (Jägermeyr et al., 2015), addressing their uncertainties is however beyond the scope of this study. Further, LPJmL does not include fossil groundwater resources which are a main source of irrigation water extraction in many regions. This also has implications for the sustainable use of blue water resources since 20% of irrigation relies on non-sustainable groundwater

withdrawals (Dalin et al., 2019). To address this gap in this study, we implemented a groundwater buffer in LPJmL in order to account for a certain level of consistent water flow below ground. For a holistic picture, it would be valuable to include information on groundwater volumes and abstractions.

## 4.3 Implications for water management and global food security

We find that 585 km³yr⁻¹ of blue water resources used violate EFRs. The EFRs were calculated in a post-process manner using

the VMF method (Pastor et al., 2014), however, further analyses could apply several EFR estimation methods and compare their results accordingly (as in Jägermeyr et al., 2017). In this study, we also do not discuss the degree to which GWS alleviation would be reduced if EFRs would be preserved internally in LPJmL. It would be furthermore beneficial to also include other sustainability criteria than the EFRs in order to account for sustainable blue water management.

In this study, irrigation was shown to have the dual effect of mitigating GWS while simultaneously increasing BWS. In

practice, this situation could be avoided by adopting more sustainable crop and water management strategies (Elsayed et al., 2025), such as enhanced green water management options (Jägermeyr et al., 2017; Gerten et al., 2020). More effective irrigation systems, for example, could limit this negative effect – further analyses could model this potential by e.g. assuming drip irrigation systems everywhere as in Jägermeyr et al. (2015). Also, water losses towards the field could be reduced by investing in better irrigation infrastructures. GWS in this study treats GWS on marginal cropland (e.g. strongly nutrient limited)

as equally important as GWS on highly productive land (with sufficient nutrient supply and pest control). By doing so, the indicator does not consider possible improvements at the crop production system, where different inputs can be managed in bundles to increase total input productivity (e.g. Yang et al., 2024). Production increases through irrigation or water-saving techniques in combination with other intensification measures could help to abandon highly water-stressed cropland elsewhere and thus nullify GWS there. However, such interactions require functional markets and alternative livelihood options for

people living there.

In this study, green and blue water stresses were calculated at a local scale, neglecting further telecoupled implications. Due to the fact that water resources and food production are increasingly interconnected through trade, water resources in rivers and groundwater aquifers might be considered local resources but increasingly gain global scope (Dalin et al., 2017; Dalin et al., 2019; Rosa et al., 2019). Yields, as well as the green and blue water resources used to produce them (referred to as virtual

water), are linking local agricultural export regions with food-dependent import regions (Allan, 2003, Dalin et al., 2017, D'Odorico et al., 2019). This highlights that the susceptibility to green and blue water scarcity extends beyond local agricultural sectors, impacting global actors and distant regions as well (Ercin et al., 2021; Rosa et al., 2019; Vallino et al., 2021). It is necessary to bridge water scarcity assessments and telecoupled flow analysis in order to get a more holistic picture of drivers and impacts at global scale (Rockström et al., 2023). Besides that, it would be valuable to conduct a set of case studies for
certain regions in order to critically reflect and corroborate this study's model-derived results.

## 5 Conclusion

This study aims at better understanding the interlinkages of GWS and BWS on agricultural areas by not only jointly mapping both stresses, but exploring where, and to what extent, GWS has driven a shift toward higher BWS. Based on the stress indicator derived from soil water supply and transpirational demand, GWS patterns are shown to vary greatly across regions
and seasons. Irrigation has alleviated GWS on 13% of the global agricultural area below the threshold of 0.2, but thereby increased BWS on 12% of the area. The analysis reveals that GWS and BWS often coincide in agricultural regions, and that 22% of the total water consumption occurs at the expense of environmental flows. This is a critical finding, as it indicates that not enough blue water resources are available to buffer GWS in a sustainable way (given the current regional distribution and efficiency of irrigation systems). The results of this study underscore the need to account for both types of water scarcity
simultaneously in strategies for sustainable agricultural water use, especially under conditions of aggravating climate change impacts.

## Code and data availability

The model code for LPJmL (version 5.9.25) used in this study is publicly available under the Creative Commons Attribution 4.0 International license at Zenodo: https://doi.org/10.5281/zenodo.16532191. The corresponding LPJmL outputs, as
well as the R scripts for the analysis and creation of the main figures are also publicly available under the Creative Commons Attribution 4.0 International license at Zenodo: https://doi.org/10.5281/zenodo.16536278.

## Author contribution

HD: conceptualization, methodology, model simulations, data analysis, writing – original draft, visualization. LA: conceptualization, methodology, writing – review and editing. SS: methodology, model simulations, writing – review and
editing. FS: methodology, writing – review and editing. JB: methodology, model simulations. CM: methodology, writing – review and editing. DG: conceptualization, methodology, writing – review and editing, supervision.



**Competing interests**

The authors declare that they have no conflict of interest.

**Acknowledgements**

We thank Yulia Suárez Bergmann for her graphic template for Figure 1. For parts of the analysis, ChatGPT has been used for coding and debugging.

**Financial support**

HD acknowledges financial support from the Heinrich-Böll foundation in the form of a PhD scholarship, as well as from the German Federal Ministry for Research and Education (BMBF) through the research project STEPSEC (grant no. 01LS2102D).
LA and JB are supported by PIK's Planetary Boundaries Science Lab. LA's research is additionally supported by BMBF, project ClimXtreme (grant no. 01LP1903D). JB's research is funded by The Grantham Foundation for the Protection of the Environment. FS is funded by the FORMAS project ReForMit.

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
