# Peer review of "Shifting water scarcities: Irrigation alleviates agricultural green water deficits while increasing blue water scarcity"

_EGUsphere, 2025_

## Author Comment (AC1)

Authors' Response to Reviews of

**Shifting water scarcities: Irrigation alleviates agricultural green water deficits while increasing blue water scarcity**

**Review #1**

The authors investigate how irrigation reshapes agricultural water stress worldwide by shifting pressure from green water scarcity (GWS) to blue water scarcity (BWS). They simulate crop water use with the global vegetation–crop model LPJmL at 0.5° daily resolution, contrasting a no-irrigation counterfactual (INO) with a limited-irrigation case (ILIM) where withdrawals are constrained to locally available blue water (including upstream inflow and reservoirs, but excluding long-distance transfers and fossil groundwater). GWS and BWS are then quantified with monthly indices.

The paper's key innovation is to treat GWS and BWS jointly, which yields clear global insights: irrigation alleviates GWS on 13% of cropland are but increases BWS by ~12%, concentrating the latter in some irrigation hotspot (e.g., in India and the Mediterranean basin). The estimates of blue water overuse are broadly consistent with published studies; however, the manuscript should explicitly state what is meant by "*water use*" (i.e., whether it refers to water consumption or to withdrawals). Moreover, it remains unclear whether renewable groundwater is explicitly included in the accounting of available blue water. Clarifying these points would strengthen the interpretation of overshoot volumes.

The manuscript is well written, the narrative is coherent, and figures, tables, and supplementary materials effectively support the analysis. Overall, this is a solid and valuable contribution; I recommend **minor revisions** focused on clarifying the treatment of renewable groundwater in the blue-water budget and related sensitivity.

We thank Dr. Lorenzo Rosa for reviewing our manuscript and for his positive evaluation of the paper. The constructive remarks and questions have been very valuable in refining our study. Below, we provide detailed responses and suggestions addressing each comment.

**Specific comments:**

** lines 39-40**: This sentence would benefit from bibliographic references; for example, Gleeson et al. (2020).

*Gleeson, T., Wang-Erlandsson, L., Porkka, M., Zipper, S. C., Jaramillo, F., Gerten, D., ... & Famiglietti, J. S. (2020). Illuminating water cycle modifications and Earth system resilience in the Anthropocene. Water Resources Research, 56(4), e2019WR024957.*

We will add the proposed reference.

**lines 115-116**: Adding (S) and (D) to water supply and atmospheric demand, respectively, would make Eq. (1) clearer and more accessible to readers, since these symbols are not defined immediately afterward.

Thank you for this comment, we will add S and D accordingly to improve the clarity of Eq. (1).

**Eq.3**: In the ratio between the scaling factor and the potential canopy conductance, the division sign (—) is missing; please check.

We will add the division sign to the equation.

**lines 145-147**: Here too, it would be better to introduce the symbols in Eq.4 beforehand. The symbols for the water uses, even if intuitive, should be restated in the text for the sake of completeness.

We will add the symbols accordingly to the sentence introducing Eq. (4).

Moreover, could you please specify what is meant by "water use" in this study? Are you working with consumption or with withdrawals? For water-balance estimates of overuse, the relevant quantity is typically identified by the water consumption, as it represents the fraction removed from the system and not locally available for reuse (hence generally smaller than withdrawals). It would help to state this explicitly in the Methods.

Thank you for this valuable comment. In the original manuscript, we did not apply a consistent definition of water use: $WU_{irr}$ was calculated with LPJmL and represents the total amount of irrigation water applied to the field. We will now recalculate $WU_{irr}$ to consistently represent irrigation water consumption as follows:

$WU_{irr} = WD_{irr} + E_{conv} - RF_{blue}$ *(Eq. 5),*

where $WD_{irr}$ represents *the applied irrigation water* which depends on the simulated irrigation system, $E_{conv}$ *the evaporative conveyance loss* and $RF_{blue}$ *the water return flow.*

The global average annual irrigation water consumption in our studied time period 2015-2019 is 1213 $km^3yr^{-1}$, very much in line with estimates calculated in other studies (e.g. Mc Dermid et al., 2024: 1195 +/- 99 $km^3yr^{-1}$ ). We assume that blue water stress will be influenced by this change and will decrease accordingly, as will be reflected in the newly calculated results.

**lines 145-162**: In these paragraphs, it is unclear why additional sectors such as livestock, electricity generation, and mining are not considered. In several regions these are as water-intensive as domestic and industrial uses. Please justify their exclusion (e.g., data gaps, scope) or discuss as a limitation.

Thank you for this important question. In our study, we used the dataset from Flörke et al. (2013) as input, which includes the water consumption of households, industry and livestock. In response to the reviews we propose to change the input and use a dataset by Huang et al. (2018) instead, which is more up-to-date and includes more sectors than Flörke et al. (2013). This dataset was created by spatially and temporally downscaling national (and U.S. state-level) sectoral water-withdrawal estimates from AQUASTAT and USGS, providing a monthly 0.5° gridded dataset for the period 1971–2010. It includes the sectors domestic, electricity generation (cooling of thermal power plants), livestock, mining, manufacturing and irrigation. For our updated analysis, we will use the data for the first five sectors as input to LPJmL (whereas consumptive water use for irrigation is computed internally).

We will update the Method section accordingly and further acknowledge data gaps and time coverage as limitations in the Discussion section and elaborate on how the inclusion of additional sectors and longer time periods could influence the results. We expect blue water

stress to (slightly) decrease because global industrial water withdrawal estimates by Flörke et al. (2013) are higher than estimates by Huang et al. (2018).

It is not specified how the Flörke et al. (1950–2010) database covers the full simulation period, particularly after 2010. Were values held constant at 2010 levels thereafter, as in Rosa & Sangiorgio (2025) and Citrini et al. (2025)? This should be stated explicitly in the Methods and acknowledged as a limitation.

Thank you for highlighting this aspect. We will adjust the Method section and describe that with the new proposed dataset, values are held constant at 2010 levels.

Please clarify why renewable groundwater inflows to grid cells are not included in the blue-water budget (they appear to be excluded in Eq. 4 and Eq. 5). If intentionally omitted, explain the rationale and, in the Discussion section, the implications for overuse estimates.

We will re-evaluate whether withdrawal from the established groundwater buffer is possible for this analysis and discuss our decision in the revised manuscript.

It would help to add a brief note on the routing module: how upstream–downstream relationships among cells are represented when computing overuse, including whether deficits are calculated locally or propagated, and any reservoir/return-flow assumptions.

The routing module represents upstream–downstream relations through the predefined river network, where discharge flows between grid cells with a constant flow velocity and via cascaded linear reservoirs (Schaphoff et al., 2018). Water deficits are calculated locally in each cell, but propagate downstream because reduced outflow from an upstream cell decreases available discharge for all downstream cells. Return flows from irrigation losses are assumed to percolate into the soil and subsequently contribute to surface runoff, which is added to the local surface water and routed downstream. Reservoirs further modify these interactions by storing, buffering, and releasing water according to their operating rules, thereby altering both local and downstream water availability. We will explain these points in the revised Method section.

**Lines 155-156**: Stenzel et al.'s approach to estimating blue-water overuse appears closely aligned with earlier assessments of unsustainable water use (e.g., Mekonnen & Hoekstra, 2016; Mekonnen & Hoekstra, 2020; Citrini et al., 2025; Rosa & Sangiorgio, 2025 and many others). It would strengthen the manuscript to situate the method explicitly within this literature (briefly clarifying similarities and differences) and, where feasible, to prioritize citations to peer-reviewed, published studies.

*Citrini, A., Sangiorgio, M., & Rosa, L. (2025). Global multi-model trends of unsustainable irrigation under climate change scenarios. Environmental Research Letters, 20(10), 104011.*

*Mekonnen, M. M., & Hoekstra, A. Y. (2016). Four billion people facing severe water scarcity. Science advances, 2(2), e1500323.*

*Mekonnen, M. M., & Hoekstra, A. Y. (2020). Blue water footprint linked to national consumption and international trade is unsustainable. Nature Food, 1(12), 792-800.*

*Rosa, L., & Sangiorgio, M. (2025). Global water gaps under future warming levels. Nature Communications, 16(1), 1192.*

Thank you very much for the suggested literature, which we were partly not yet aware of (2025 publications). We will situate our method within the existing studies and discuss

similarities and differences in the approaches and results in the restructured discussion section. We also add the baseline (2001-2010) result of Citrini et al. (2025) / Rosa and Sangiorgio (2025) to Table 1 as a comparative reference for validation.

**Figure2**: The figure is excellent. One suggestion would be to revisit the colorbar and its tick labels. Because the text consistently refers to thresholds at 0.2–0.4–0.6–0.8–1.0, harmonizing the colorbar bins/ticks with those intervals (similar to the style used in Fig. S11) would improve readability and make cross-references more immediate. If a full reclassification is not desired, introducing color breakpoints at those thresholds would still create a clear visual link to the classes cited in the text. Please take this as an optional refinement to consider.

It would also help to state explicitly in the caption that the colorbar applies to all panels in the figure (as you did in the caption of Figure 4).

Thank you very much for this comment, we will adapt the proposed changes in order to improve the readability of Figure 2.

**lines 181-184**: Another enhancement to consider is splitting Figure 2 into five panels (2a–e) instead of two, so the manuscript can reference each component explicitly. For example: "*The green water stress patterns show a high seasonal variability over the year due to changing weather conditions but also season-specific growing seasons (**Fig. 2b-e**). Europe and North America do experience less (or even no) GWS during the winter months since the water demand of the crops grown during that time is very low (**Fig. 2b**). During the summer months (**Fig. 2d**), however, especially southern regions in Europe and the western US are highly green water stressed (GWS >0.4). Large regions in Brazil change into GWS hotspots from June to November where pulses, rapeseed and sugarcane are especially green water stressed (**Fig. 2d-e**). India, by contrast, does not experience high GWS from June to November (**Fig. 2d-e**), when most crops are grown.*"

We agree with the proposed splitting of Figure 2 and will adjust the figure accordingly.

**Figure 5**: The figure appears very similar to the patterns reported by Citrini et al. (2020) for the 2001–2010 baseline (see their Supplementary Fig. 2a). It would strengthen the paper to briefly position your map against those results - highlighting key consistencies or divergences and the likely reasons in the Discussion section. This seems especially pertinent if your overuse metric is restricted to surface-water resources, whereas Citrini et al. used a source-agnostic approach (i.e., not distinguishing whether scarcity arises from groundwater or surface water).

*Citrini, A., Sangiorgio, M., & Rosa, L. (2025). Global multi-model trends of unsustainable irrigation under climate change scenarios. Environmental Research Letters, 20(10), 104011.*

Thank you very much for this suggestion, we will include the study of Citrini et al. (2025) in our paper and discuss the similarities and differences accordingly in the revised Discussion section.

**Discussion section**: The Discussion would benefit from engaging with the most recent literature published this year, e.g., Rosa & He (2025) for GWS, and Rosa & Sangiorgio (2025) together with Citrini et al. (2025) for blue-water overuse. Situating your findings alongside these studies (noting agreements, differences, and methodological nuances) would further strengthen the contribution. Could you clarify what drives the difference between the estimates (585 km$^3$yr$^{-1}$ vs ~460 km$^3$yr$^{-1}$)? Is the difference mainly due to (i) restricting bluewater availability to surface sources, (ii) using withdrawals rather than consumptive use, or (iii) other methodological choices (e.g., treatment of return flows, reservoirs, EFRs, routing, baseline years)?

In addition, the sentence at lines 302–304 should be supported with appropriate references.

*Citrini, A., Sangiorgio, M., & Rosa, L. (2025). Global multi-model trends of unsustainable irrigation under climate change scenarios. Environmental Research Letters, 20(10), 104011.*

*Rosa, L., & He, L. (2025). Global multi-model projections of green water scarcity risks in rainfed agriculture under 1.5° C and 3° C warming. Agricultural Water Management, 314, 109519.*

*Rosa, L., & Sangiorgio, M. (2025). Global water gaps under future warming levels. Nature Communications, 16(1), 1192.*

We will improve our discussion by incorporating this very recent literature on blue water consumption and overuse. We anticipate that our estimate of 585 km³ yr⁻¹ of blue water from non-sustainable surface water resources will likely decrease following the recalculation of irrigation water consumption, and we will elaborate on this accordingly in the revised manuscript.

**lines 314-315**: It would be helpful to briefly outline how such these new case studies would be implemented. For example, would higher spatial and/or temporal resolution be required? If so, please indicate the additional data needs (e.g., finer-resolution water use, irrigation, hydrologic, and management datasets) and whether the availability of such data currently constrains feasibility.

Thank you for this comment. Indeed, additional datasets with higher spatial and temporal resolution—particularly for irrigation and management—would be highly valuable. The case studies mentioned in this section, however, were intended to take a more qualitative approach, relying on interviews and bottom-up analyses together with actors affected by water scarcity to better understand local impacts and potential solutions. Since global modeling studies like ours depend on large gridded datasets, such qualitative insights would help validate whether the changes and effects in water use and stress that we identify are also observed by local actors. A more interdisciplinary approach would therefore be highly beneficial, and we will shortly elaborate on this in the discussion section.

**Reference list**: Just a minor formatting note, likely governed by the journal's template rather than the authors: applying a hanging indent to the paragraph/list would make the items much easier to scan and review.

Thank you for this suggestion, we will apply a hanging indent to the reference list in the revised manuscript.

Thank you for the opportunity to revise the study. Again, this is great work (congratulations!) and minor clarificatory revisions are requested.

Lorenzo Rosa

**References**

Citrini, A., Sangiorgio, M. and Rosa, L.: Global Multi-Model Trends of Unsustainable Irrigation under Climate Change Scenarios, ERL, 20, 10, 104011, https://doi.org/10.1088/1748-9326/adfcee, 2025.

Flörke, M., Kynast, E., Bärlund, I., Eisner, S., Wimmer, F. and Alcamo, J.: Domestic and industrial water uses of the past 60 years as a mirror of socio-economic development: A global simulation study, Global Env. Change, 23, 1, 144-156, https://doi.org/10.1016/j.gloenvcha.2012.10.018, 2013.

Huang, Z., Hejazi, M., Li, X., Tangl, Q., Vernon, C., Leng, G., Liu, Y., Döll, P., Eisner, S., Gerten, D., Hanasaki, N. and Wada, Y.: Reconstruction of Global Gridded Monthly Sectoral Water Withdrawals for 1971–2010 and Analysis of Their Spatiotemporal Patterns. HESS, 22, 4, 2117–33, https://doi.org/10.5194/hess-22-2117-2018, 2018.

McDermid, S., Nocco, M., Lawston-Parker, P., Keune, J., Pokhrel, Y., Jain, M., Jägermeyr, J., Brocca, L., Massari, C., Jones, A.D., Vahmani, P., Thiery, W., Yao, Y., Bell, A., Chen, L., Dorigo, W., Hanasaki, N., Jasechko, S., Lo, M., Mahmood, R., Mishra, V., Mueller, N.D., Niyogi, D., Rabin, S.S., Sloat, L., Wada, Y., Zappa, L., Chen, F., Cook, B.I., Kim, H., Lombardozzi, D., Polcher, J., Ryu, D., Santanello, J., Satoh, Y., Seneviratne, S., Singh, D. and Yokohata, T.: Irrigation in the Earth System, Nat Rev. Earth & Env., 4, 7, 435–53, https://doi.org/10.1038/s43017-023-00438-5, 2024.

Rosa, L.,and Sangiorgio. M.: Global Water Gaps under Future Warming Levels, Nat. Comm., 16, 1, 1192, https://doi.org/10.1038/s41467-025-56517-2, 2025.

Schaphoff, S., Von Bloh, W., Rammig, A., Thonicke, K., Biemans, H., Forkel, M., Gerten, D., Heinke, J., Jägermeyr, J., Knauer, J., Langerwisch, F., Lucht, W., Müller, C., Rolinski, S., and Waha, K.: LPJmL4 – a dynamic global vegetation model with managed land – Part 1: Model description, Geosci. Model Dev., 11, 1343–1375, https://doi.org/10.5194/gmd-11-1343-2018, 2018.

---

## Author Comment (AC2)

**Authors' Response to Reviews of**

**Shifting water scarcities: Irrigation alleviates agricultural green water deficits while increasing blue water scarcity**

**Review #2**

**General comments**

This paper sheds light on the interlinkages between green (GWS) and blue water scarcity (BWS) as simulated by the global gridded vegetation model LPJmL. The results are presented for the 2015–2019 period with a sufficient level of detail and supporting visualisations. The main finding suggests that irrigation helps humanity to alleviate GWS impacts on crops, but this comes at the cost of increasing BWS. As such, this paper contributes to the growing body of literature highlighting the need to analyse both green and blue water use simultaneously when addressing water stress impacts on crop production worldwide.

However, as always, there are multiple things the authors can improve upon. My main concern lies in the definition of GWS and how robust the results are. Choosing 0.2 as the main GWS threshold seems rather arbitrary and leaves me wondering how different the results would be if the authors had chosen 0.1 or 0.3 instead (I explain below in detail). Adding to this, there is no clear section on the validation of the underlying global estimates by LPJmL. The full list of potential issues to address is provided below.

We would like to thank Reviewer 2 for the thorough and detailed review. We greatly appreciate the constructive feedback and insightful questions, which will significantly contribute to improving the quality of the paper. We provide below our point-by-point responses together with our proposals for implementing the suggested improvements.

**Specific comments:**

Regarding the definition of GWS:

1. Authors use the 0.2 GWS threshold as earlier applied by Rosa et al. (e.g. https://doi.org/10.1093/pnasnexus/pgad117). The assumption is that such a level of GWS would force farmers to irrigate to avoid large yield declines. However, this assumption is rather arbitrary and does not rely on actual crop yield decline simulations (as you also acknowledge in L245-246). As far as I know, LPJmL is capable of simulating crop yields, so why not using those to define reasonable thresholds per crop functional type (CFT)? I can imagine that some crops indeed can have substantial yield declines at 0.2 GWS, but others might not be so sensitive.

We thank the reviewer for this very valuable comment. We acknowledge this limitation, and to address it, we propose a revised approach that derives green water stress thresholds directly from simulated yield responses – as described in the summary response to all reviewers. With this approach we intend to ensure a closer and more meaningful linkage between green water stress levels and their agricultural impacts. Our preliminary findings

indicate differences in the CFT yield response with regard to GWS: while many CFTs exhibit yield losses of approximately 20–30% at a green water stress level of 0.2 under rainfed conditions, certain CFTs (e.g. pulses and temperate cereals) experience considerably higher losses under similar stress conditions.

We will describe this enhanced methodological framework in detail in the Methods section and compare the resulting CFT-specific thresholds, as well as the associated stressed agricultural areas, with the previously applied uniform thresholds of 0.2 and 0.4. Finally, the Discussion section will address the implications and limitations of this refined approach.

2. Connected to the above, consider reframing the analysis from only two main GWS categories (0.2-0.4 and 0.4-1.0) to sequential steps of 0.2 or 0.25 to have graduality in GWS exposure (e.g. no GWS, minor, moderate, severe). Authors already do this for Fig. 4, for example, why not in the main text and Table 1 too?

Thank you very much for this proposal. Connected to the adapted new approach, we will also reframe the GWS thresholds/categories to sequential steps of 0.2, in order to show a graduality in the GWS exposure. We will define the thresholds as following:

0-20 % yield decline   = low GWS
20-40% yield decline   = moderate GWS
40-60% yield decline   = high GWS
60-80% yield decline   = severe GWS
80-100% yield decline = extreme GWS

We will adjust the text accordingly and include a new figure that illustrates the CFT-specific green water stressed areas.

3. Authors define green water as "*the plant available rainwater held in soils which sustains the growth of crops and pastures*", and then they also state that "*irrigation is shown to alleviate green water stress*". I find this a bit confusing. If green water = rainwater, then how come irrigation can alleviate GWS? The total water stress (from insufficient green+blue water supply) can indeed be reduced by supplementary blue water supply via irrigation, but green water scarce areas remain green water scarce by definition of what green water is. There is still not enough rainwater in the soil even after adding irrigation, and this statement would hold unless the precipitation patterns and/or soil texture change. Therefore, please consider rephrasing the parts where such confusion can occur or perhaps use total water stress (or some other general term) to make the distinctions clearer.

Thank you very much for pointing out this inconsistency. We propose adjusting the green water definition to: "plant available soil moisture from rainfall and snow melt which sustains the growth of crops and pastures." We further acknowledge that blue water cannot alleviate GWS, but that it can alleviate/compensate for plant water stress if GWS is high. With irrigation, GWS remains unchanged but the stress level that plants experience is alleviated by supplementing blue water. We will rephrase the parts accordingly.

Regarding the definition of BWS:

1. Is there a particular reason why water use (as I understand, that means withdrawals) as opposed to water consumption (actual water volume removed from a catchment) is used? To my knowledge, most recent BWS studies use the latter since a large part of the withdrawals stays within the same catchment, and thus, should not contribute to BWS in a long term.

Thank you for this valuable comment. In the original manuscript, we did not apply a consistent definition of water use: $WU_{irr}$ was calculated with LPJmL and represents the applied irrigation water at the field whereas $WU_{dom}$ and $WU_{ind}$ represent water consumption. In response to the feedback, we will recalculate $WU_{irr}$ to consistently represent irrigation water consumption as follows:

*$WU_{irr} = WD_{irr} + E_{conv} - RF_{blue}$ (Eq. 5),*

where *$WD_{irr}$* represents *the applied irrigation water* which depends on the simulated irrigation system, *$E_{conv}$ the evaporative conveyance loss* and *$RF_{blue}$ the water return flow.*

The global average annual irrigation water consumption in our time period 2015-2019 is approximately 1213 $km^3yr^{-1}$, which is comparable to estimates calculated in other studies (e.g. Mc Dermid et al., 2024: 1195 +/- 99 $km^3yr^{-1}$). We assume that the blue water stress will be highly influenced by this change and will decrease accordingly.

2. It is not clear where water use estimates for domestic and industrial use are taken from, and whether they account for monthly variability. Please add more details.
3. It is not clear whether the authors consider groundwater withdrawals as well as livestock, electricity generation, and mining water supplies. Please elaborate.

Thank you for this important question. In our study, we used the dataset from Flörke et al. (2013) as input, which includes the water consumption of households, industry and livestock. In response to the reviews we propose to change the input and use a dataset by Huang et al. (2018) instead which is more up-to-date and includes more sectors. This dataset was created by spatially and temporally downscaling national (and U.S. state-level) sectoral water-withdrawal estimates from AQUASTAT and USGS, providing a monthly 0.5° gridded dataset for the period 1971–2010. It includes the sectors domestic, electricity generation (cooling of thermal power plants), livestock, mining, manufacturing and irrigation. For our updated analysis, we will use the data for the first five sectors as input to LPJmL (whereas consumptive water use for irrigation is computed internally).

We will update the Method section accordingly and further acknowledge data gaps and time coverage as limitations in the Discussion section and elaborate on how the inclusion of additional sectors and longer time periods could influence the results. We expect blue water stress to (slightly) decrease because global industrial water withdrawal estimates by Flörke et al. (2013) are higher than estimates of Huang et al. (2018).

Regarding the model setup:

1. I find the description of LPJmL inputs in the main text insufficient. This study relies entirely on results from this model, so, as a reader, I would expect to have a very

detailed description of input data and main assumptions in Section 2.2, and validation, limitations, and uncertainties in Section 4. Please try to elaborate. I provide several suggestions below.

We acknowledge that the description of the model setup as well as of the input data can be enhanced. We will improve section 2.2 accordingly by describing the LandInG toolbox as well as the input data in more detail.

2. In addressing 3a, I would recommend moving Table S1 into the main text, as a reader needs to see the main input data sources and their description. Also, consider adding a column with spatial resolution next to "Time period".

Thank you for this suggestion which we will adopt.

3. When describing the land use and CFTs, please explain:
- where the LandInG toolbox gets rainfed/irrigated crop maps from, and whether those maps cover spatiotemporal changes

We thank the reviewer for this relevant question. The LandInG toolbox derives rainfed and irrigated crop maps from a combination of country-level and gridded datasets, including FAOSTAT, MIRCA2000, AQUASTAT, MON, RAM, and HYDE. Country-level datasets provide information on crop-specific harvested areas and changes over time, while gridded datasets provide spatial detail or temporal dynamics but may lack crop-specific information. LandInG integrates these sources to create a harmonized, gridded dataset at 0.5 resolution, covering 1500–2017, by disaggregating country-level data to grid cells and resolving inconsistencies between datasets. This processing allows the toolbox to capture spatiotemporal changes in rainfed and irrigated crop distributions while maintaining consistency with crop-specific and irrigation-specific information available at coarser, e.g. national, resolution. We will explain the LandInG toolbox in more detail in section 2.2 accordingly.

- where crop calendars are from, and whether they are static or dynamic

We use the GGCMI Phase 3 crop calendar described in Jägermeyr et al. (2021) which is static in sowing dates and variety parameters, meaning that seasons start the same day in each year, but the length of the growing season is variable according to temperature variations. We will add these details to the revised Method section.

- what share of primary crops from the crop list provided by FAOSTAT is covered (otherwise, it is not clear what your scope is), see L85-87. Also mention if perennial crops are covered.

We adopt 100% of the cropland area from the HYDE database, which corresponds to 100% of the FAO cropland. However, a significant share of this area may be classified as the category "others", which aggregates all crops not parameterised specifically as CFTs. Many FAOSTAT primary crops fall under this "others" category, e.g. all FAO primary crops of the category "Seeds & Oils" except sunflower and rapeseed, and also many vegetables. "Others" include

also some perennial crops, e.g. coffee, cocoa and tea. The 12 parameterised CFTs cover ≈ 60% of the global agricultural area while "others" cover ≈ 40%. We will provide more detail on these aspects in the revised Method section, and mention in the Discussion how the less specific parameterisations may influence results.

4. When describing the soil, please specify whether the root zone is dynamic or kept static over the growing season, mention how many soil layers are simulated, and whether the shallow groundwater is considered (since capillary rise can support some rainfed crops).

In our model setup, the root zone is kept static over the growing season. The soil column is represented by six layers, of which five are hydrologically active, with depths of 0.2, 0.3, 0.5, 1.0, and 1.0 m (see Schaphoff et al., 2018). While shallow groundwater is implicitly considered through the implementation of baseflow, capillary rise is not represented. We will incorporate these clarifications into the revised manuscript.

5. When describing the irrigation, clearly state what irrigation systems are considered (furrow, sprinkler, drip), how and when irrigation is applied, and whether conveyance losses are simulated.

We thank the reviewer for this helpful comment. In our study, each country is assigned a share of the three irrigation systems (furrow, sprinkler, and drip) following the approach of Jägermeyr et al. (2015). These national-level shares are further disaggregated to the grid cell and crop functional type (CFT) level using a decision-tree approach that accounts for crop- and soil-specific suitability. Irrigation is applied according to these distributions, which are updated annually. Conveyance losses are simulated and depend on the irrigation systems as well as soil conditions. For pressurized systems (sprinkler and drip irrigation), the conveyance efficiency is set to 0:95 while for surface irrigation conveyance efficiency is connected to different soil saturated hydraulic conductivities (see Jägermeyr et al., 2015 (Table 1)).

Regarding validation, limitations, and uncertainties:

1. As mentioned earlier, I would expect to see a sub-section on validation, limitations, and uncertainties in Section 4. The presently provided comparisons mainly look at GWS and BWS in terms of hectares. However, the underlying gridded estimates of green and blue ET, as well as sectoral blue water demand, are not properly validated. Please add such comparisons (global total and/or gridded levels), maybe a brief summary for the main text and an elaborated version in SI. Otherwise, it is difficult to judge whether the LPJmL model outputs are reliable before even diving into GWS/BWS analysis.

Thank you very much for this valuable suggestion, we agree that a sub-section on validation, limitation and uncertainties in the Discussion section will be beneficial, so we will add it.

2. For green and blue ET (i.e. water consumption) estimates, you can have a look at https://doi.org/10.1038/s41597-020-00612-0 and https://doi.org/10.1038/s41597-024-03051-3, but there also might be more studies published recently.

We propose to expand the global overview table in the manuscript (Table 1) to include additional hydrological variables, such as evapotranspiration. We will also check spatial patterns and discuss them in comparison with findings from other studies (e.g. those suggested here). We already validated the evaporation rates of this study with evaporation rates measured at eddy flux towers (in the Supplementary Material) and will point to this more clearly.

3. Sectoral blue water use can be obtained from global hydrological models provided by ISIMIP as well as AQUASTAT

The sectoral water use input dataset of Huang et al. (2018) that we propose to use instead of the input dataset of Flörke et al. (2013) is supported by observational studies and well evaluated through uncertainty analyses (Huang et al., 2018). We will discuss these results in the revised manuscript.

4. Lastly, I would recommend having an additional sub-section describing 1) main limitations and uncertainties (coming from input data, LPJmL-specific ones, etc.) and 2) discussing how those limitations and uncertainties affect the reliability of the results. For example, how BWS could change if the authors used a water consumption-based approach instead of water withdrawals or how different EFR methods could affect the results. No need to run additional simulations to provide an uncertainty range, but it is always a good practice to be open about the weaknesses of the selected methodology, so the follow-up studies can address those.

Thank you for this helpful suggestion. We will add a dedicated sub-section in the revised manuscript outlining the main limitations and uncertainties, including those related to input data and LPJmL-specific assumptions. We will also discuss how these limitations may affect the reliability of our results.

**Technical corrections:**

1. L13: Authors define GWS as "soil moisture limitation on crop growth" in the abstract without mentioning that it only concerns rainwater.

As mentioned above, we propose to adapt the definition of GWS and will adjust it here accordingly.

2. L30: Please consider using more recent publications to support your statement.

We added two more recent references: Mialyk et al., 2024 and Chukalla et al., 2025.

3. L32: Authors mention green water scarcity without defining what it is.

We will add a definition here.

4. L36: Rainfed croplands can also get blue water via capillary rise, see https://doi.org/10.1088/1748-9326/ad78e9 . Also, correct the citation for (X. Liu et al., 2022) to fit journal guidelines (check through the manuscript for other instances).

We will include capillary rise as a water source for rainfed agriculture.

In our manuscript, we cite two authors named Liu who published in the same year [2022], which is why we distinguish them in the citation using initials.

5. L47: Maybe use EFR instead of "environmental water requirements" since you already introduced it a few lines earlier.

Will do.

6. L62: Define what the LPJmL abbreviation is.

Will do.

7. L98-99: It is not clear why running the 1901-2019 period with a 3500-year spin-up is needed. The analysis is mainly for 2015-2019. So why starting in 1901, and why so many spin-up years? Please elaborate.

The 3,500-year spin-up period has been applied for the natural vegetation in order to bring the plant functional type (PFT) distributions as well as the carbon and nitrogen pools into a dynamic pre-industrial equilibrium. This spin-up is generally needed so that pool sizes and simulated dynamics are not affected by model drift or inconsistencies between an initialization of pools and internally computed rates. The subsequent land-use spin-up (1500-2014), allows the model to incorporate historical land-use changes. From 1901 onwards, we have a transient climate input (before it was shuffled climate input from 1901-1930), which is needed for including the effect of historical land use and climate on agricultural soil properties, amongst others. After this, the analysis period (2015-2019) starts. We will explain this in more detail in the Supplementary Material.

8. L112: Please explain what 0.01 rate is (units? physical meaning?) and how it was selected.

We adopted the baseflow rate from Döll et al. (2003), it has the unit 1/d, which means that 1% of the current buffer volume is released as baseflow each day.

9. L120: Based on what 8mm/d as max is defined?

In our study, we used the revised maximum daily transpiration rate ($E_{max}$ = 8 mm day$^{-1}$) from Fader et al. (2010). We will include this reference in the revised manuscript.

10. L126-127: Please provide references to the coefficient and scaling factor (or explain how you calculated them)

Thank you for this comment, we will add references for both variables: Huntingford and Monteith (1998) for the scaling factor (from which we took the average value) and Priestley and Taylor (1972) for the coefficient.

    11. Please add units to Eq. 2 and 3.

Will do.

    12. How do you calculate annual representative GWS values based on monthly numbers? I find it a bit unclear. Perhaps add an equation to better explain L137-139.

To derive annual GWS values from monthly outputs, we calculated the mean across all months. We acknowledge that this approach smooths out seasonal variations, which is why we present seasonal maps in Figure 2.

    13. L148: WUdom, WUind, and WUirr are not defined.

We will define the water use input data accordingly in the revised manuscript.

    14. L187-188: consider adding a sentence with additional global GWS estimates for each year or simply provide a range (min, max) during 2015-2019.

Will do.

    15. In Table 1, the total area for ILIM under GWS>0.2 is larger for rainfed crops (426+213 Mha) than for all crops combined (431+161 Mha). Please double-check.

Thank you for raising this point. This happened due to an inconsistency in the calculation as averaged values of rainfed and irrigated cropland were summed up. We will calculate the areas differentiated into rainfed and irrigated in the revised manuscript.

    16. L197-198: If 37% of all croplands are under GWS>0.2, then logically the remaining lands (63%) are under GWS<0.2. However, the authors say it is only 13%. Please double-check.

50% of the cropland experience GWS > 0.2 with the INO scenario (no irrigation), 37% with the ILIM scenario (limited irrigation). The 13% therefore state that "13% of the global cropland area experiences GWS <0.2 due to the alleviating effect of irrigation" (line 198).

17. L202-204: From the text, it is not immediately clear what the difference is with L195-197. After a few minutes, I finally realised that it is only about irrigated grid cells here (rainfed excluded). Please make it clear for the reader.

Thank you for this comment, we will rephrase the sentence so that it is directly clear that these values refer to grid cells with irrigation.

18. L204: If indeed only irrigated cells are analysed, please explain why there are still large GWS areas remaining (once you switch on irrigation). Is it because some irrigated cells cannot have access to a sufficient blue water supply, so the crops remain under water deficit? If so, how realistic is this assumption? I would expect that such underirrigation is not that common globally, since farmers can always dig into groundwater reserves or simply ignore EFR (perhaps mention this in the sub-section on limitations and uncertainties).

Thank you for this remark. Figure 4b presents results for all agricultural areas within grid cells that contain irrigation, meaning both rainfed and irrigated cropland. Due to that fact, it is possible that even grid cells that have irrigation contain higher levels of GWS. We chose this approach to ensure comparability with the BWS index, which is not CFT-specific and refers to the entire grid-cell area. We will clarify this more explicitly in the revised Results section.

19. L296: Consider providing a few examples of green water management options.

Thank you for this suggestion, we will include examples of green water management options in the revised manuscript.

As you can see, I took a great responsibility of being the "*annoying reviewer #2*". Hope my constructive criticism will be helpful in improving the quality of this paper, which I am truly looking forward to reviewing in the next round of revisions. Good luck!

**References**

Chukalla, A. D., Mekonnen, M. M., Gunathilake, D., Wolkeba, F. T., Gunasekara, B. and Vanham, D.: Global Spatially Explicit Crop Water Consumption Shows an Overall Increase of 9% for 46 Agricultural Crops from 2010 to 2020, Nature Food, 6, 10, 983–94, https://doi.org/10.1038/s43016-025-01231-x, 2025.

Döll, P., Kaspar, F. and Lehner, B.: A Global Hydrological Model for Deriving Water Availability Indicators: Model Tuning and Validation, Journal of Hydrology, 270, 1–2, 105–34, https://doi.org/10.1016/S0022-1694(02)00283-4, 2003.

Fader, M., Rost, S., Müller, C., Bondeau, A. and Gerten, D.: Virtual Water Content of Temperate Cereals and Maize: Present and Potential Future Patterns, Journal of Hydrology, 384, 3–4, 218–31, https://doi.org/10.1016/j.jhydrol.2009.12.011, 2010.

Flörke, M., Kynast, E., Bärlund, I., Eisner, S., Wimmer, F. and Alcamo, J.: Domestic and industrial water uses of the past 60 years as a mirror of socio-economic development: A global simulation study, Global Env. Change, 23, 1, 144-156, https://doi.org/10.1016/j.gloenvcha.2012.10.018, 2013.

Huang, Z., Hejazi, M., Li, X., Tangl, Q., Vernon, C., Leng, G., Liu, Y., Döll, P., Eisner, S., Gerten, D., Hanasaki, N. and Wada, Y.: Reconstruction of Global Gridded Monthly Sectoral Water Withdrawals for 1971–2010 and Analysis of Their Spatiotemporal Patterns. HESS, 22, 4, 2117–33, https://doi.org/10.5194/hess-22-2117-2018, 2018.

Huntingford, C., and Monteith, J. L.: The Behaviour of a Mixed-Layer Model of the Convective Boundary Layer Coupled to a Big Leaf Model of Surface Energy Partitioning, Boundary-Layer Meteorology, 88, 1, 87–101, https://doi.org/10.1023/A:1001110819090, 1998.

Jägermeyr, J., Gerten, D., Heinke, J., Schaphoff, S., Kummu, M. and Lucht, W.: Water savings potentials of irrigation systems: global simulation of processes and linkages, Hydrology and Earth System Sciences, 19, 3073–3091, https://doi.org/10.5194/hess-19-3073-2015, 2015.

Jägermeyr, J., Müller, C., Ruane, A.C., Elliott, J., Balkovic, J., Castillo, O., Faye, B., Foster, I., Folberth, C., Franke, J.A., Fuchs, K., Guarin, J.R., Heinke, J., Hoogenboom, G., Iizumi, T., Jain, A.K., Kelly, D., Khabarov, N., Lange, S., Lin, T.-S., Liu, W., Mialyk, O., Minoli, S., Moyer, E.J., Okada, M., Phillips, M., Porter, C., Rabin, S.S., Scheer, C., Schneider, J.M., Schyns, J.F., Skalsky, R., Smerald, A., Stella, T., Stephens, H., Webber, H., Zabel, F., Rosenzweig, C.: Climate Impacts on Global Agriculture Emerge Earlier in New Generation of Climate and Crop Models, Nature Food, 2, 11, 873–85, https://doi.org/10.1038/s43016-021-00400-y, 2021.

McDermid, S., Nocco, M., Lawston-Parker, P., Keune, J., Pokhrel, Y., Jain, M., Jägermeyr, J., Brocca, L., Massari, C., Jones, A.D., Vahmani, P., Thiery, W., Yao, Y., Bell, A., Chen, L., Dorigo, W., Hanasaki, N., Jasechko, S., Lo, M., Mahmood, R., Mishra, V., Mueller, N.D., Niyogi, D., Rabin, S.S., Sloat, L., Wada, Y., Zappa, L., Chen, F., Cook, B.I., Kim, H., Lombardozzi, D., Polcher, J., Ryu, D., Santanello, J., Satoh, Y., Seneviratne, S., Singh, D. and Yokohata, T.: Irrigation in the Earth System, Nat Rev. Earth & Env., 4, 7, 435–53, https://doi.org/10.1038/s43017-023-00438-5, 2024.

Mialyk, O., Booij, M. J., Schyns, J. F. and Berger, M.: Evolution of Global Water Footprints of Crop Production in 1990–2019, ERL,19, 11, 114015, https://doi.org/10.1088/1748-9326/ad78e9, 2024.

Priestley, C. H. B. and Taylor, R., J.: On the Assessment of Surface Heat Flux and Evaporation Using Large-Scale Parameters, Monthly Weather Review, 100, 2, 81–92, https://doi.org/10.1175/1520-0493(1972)100<0081:OTAOSH>2.3.CO;2, 1972.

Schaphoff, S., Von Bloh, W., Rammig, A., Thonicke, K., Biemans, H., Forkel, M., Gerten, D., Heinke, J., Jägermeyr, J., Knauer, J., Langerwisch, F., Lucht, W., Müller, C., Rolinski, S., and Waha, K.: LPJmL4 – a dynamic global vegetation model with managed land – Part 1: Model description, Geosci. Model Dev., 11, 1343–1375, https://doi.org/10.5194/gmd-11-1343-2018, 2018.

---

## Author Comment (AC3)

Authors' Response to Reviews of

**Shifting water scarcities: Irrigation alleviates agricultural green water deficits while increasing blue water scarcity**
* * *
**Review #3**

The study leverages two indicators: green water scarcity (GWS) and blue water scarcity (BWS), to quantify and examine the interconnectedness of green and blue water resources for irrigation. They show that while irrigation alleviates GWS, it simultaneously exacerbates BWS at the expense of environmental flow supply. The paper successfully addresses the research gap with clear narration and consistent structure throughout. Given that the results hinge on several salient parameters ($E_{max}$, threshold values), I recommend that authors consider conducting a sensitivity analysis to strengthen their analysis and to enable both themselves and the readers to better gauge the robustness of the results.

We cordially thank Reviewer 3 for taking the time to review our manuscript and for providing valuable and constructive feedback. Below, we provide detailed responses to all comments, including proposals to address the suggested improvements.

Major comments:

- There is still room to further improve the transparency of the modelling process. This could include making assumptions more explicit and providing justification or citations for modelling decisions so that readers can trace and assess their epistemic quality/reasoning. Such steps would also support future scholars in replicating the work based solely on the provided documentation. Please see the following details:
  - **L. 90**: Could you expand on the parameterisation of irrigation efficiency? Since CFTs may be suitable to more than one irrigation method (see Jagermeyr et al 2015), how did you determine which method to apply? Was the efficiency value based on an area- or grid-cell-weighted average?

Thank you very much for raising this point. We acknowledge that the description of the modelling process and decisions can be improved. In the revised manuscript, we will explain steps more clearly, and justify our methodological choices in greater detail.

To determine which irrigation method to apply to each CFT, decision rules have been developed by Jägermeyr et al., 2015 (based on Brouwer et al., 1988; Sauer et al., 2010; Fischer, 2012). These rules specify the suitability of surface, sprinkler, and drip systems for each CFT (summarized in Jägermeyr et al., 2015, Table 2). For all CFTs that may be suitable for more than one irrigation method, a structured allocation algorithm was applied: First, for each country, all grid cells with drip-suitable CFTs were identified, and CFT fractions were randomly sampled until the national target area for drip irrigation was met. This procedure was repeated 1000 times, and the iteration best matching the national target was selected. Second, sprinkler irrigation was assigned following the same logic, and the remaining irrigated area was allocated to surface systems (see Jägermeyr et al., 2015 (Supplementary Material)). In this way, the method selection is not based solely on CFT suitability but also constrained by

observed national irrigation system shares. As a result, each CFT in each grid cell is allocated to one of the three irrigation systems.

The parameterisation of irrigation efficiency depends on the irrigation systems as well as soil conditions. For pressurized systems (sprinkler and drip irrigation), the conveyance efficiency is set to 0:95 while for surface irrigation conveyance efficiency is connected to different soil saturated hydraulic conductivities (see Jägermeyr et al., 2015 (Table 1)). The irrigation efficiency values are based on the area-weighted shares of irrigation systems at the grid-cell level.

- **L. 120**: Could you provide a rationale for using $E_{max}$ = 8 mm/day? Earlier studies suggest lower values (e.g., Gerten et al 2004 report 5-7 mm/day, while Rost et al, 2008 used 5 mm/day). If more recent studies support your chosen value, it would be helpful to cite them.

In our study, we used the revised maximum daily transpiration rate ($E_{max}$ = 8 mm day$^{-1}$) from Fader et al. (2010). We will include this reference in the revised manuscript and seek additional supporting literature.

- **L. 129**: The statement "*If S > D, we set S = D to ensure the result remains within [0,1]*" could benefit from clearer justification. For example: "When S>D, we set S = D, because plants cannot take up more water than their transpiration demand allows." This framing preserves the rationale while adding a physiological explanation.

Thank you very much for this suggestion which we adopted.

- **L. 143**: Could you elaborate on the rationale for selecting 0.4 as the threshold to delineate highly water-scarce conditions? Since this assumption conditions the characterisation of scarcity across regions, you may consider conducting a sensitivity analysis to assess the soundness of your results to alternative threshold values.

We acknowledge that the choice of the threshold levels 0.2 and 0.4 was not robustly tested in the original manuscript. With the new threshold approach - building on actual yield declines of specific CFTs due to green water stress (as described in the general response to all reviewers) - we propose reframing the GWS categories to sequential steps of 0.2, in order to show a graduality in the GWS exposure. We will define the categories as following:

0-20 % yield decline    = low GWS
20-40% yield decline   = moderate GWS
40-60% yield decline   = high GWS
60-80% yield decline   = severe GWS
80-100% yield decline = extreme GWS

We will adjust the text accordingly.

- **L. 256:** It may be helpful to mention the non-representation of multiple cropping already in the methodology, rather than only in the discussion, as this limitation (as you argued) could underestimate your results.

Thank you for the suggestion, we will adjust the Method section accordingly.

- Given that the results are highly contingent on several parameterisations, it may be beneficial for the authors to conduct a sensitivity analysis to strengthen their conclusions and to allow stakeholders to better gauge the robustness of the findings. For example, sensitivity runs on influential elements such as $E_{max}$ values (5-8 mm/day) and different thresholds (0.2, 0.3, 0.4, 0.6) may be informative and also support your choice of values.

In the revised analysis, we will use sequential steps as GWS thresholds and we will provide a sensitivity analysis showing how the threshold definition can influence the results of the area affected. We will furthermore compare the new threshold results to our original assumptions (0.2 and 0.4).

We acknowledge that $E_{max}$ values usually range between 5-8 mm (as noted above). We already validated the evaporation rates of this study with evaporation rates measured at eddy flux towers (in the Supplementary Material) and will point to this more clearly. In addition, we propose to include global ET values in Table 1 to facilitate better comparison.

Minor comments:

- For consistency with previous LPJmL studies, please consider standardising the equation symbols:
  - **L. 124:** $a_m$ should be written as $\alpha_m$
  - **L. 124:** adjust the fraction form of $g_m/g_c$ for clarity

Will do.

- The authors may consider aligning the numbering of supplementary materials with the order in which they are first referenced in the manuscript. Additionally, there are several information and figures included in the supplementary material that are not explicitly mentioned in the main text. Without such references, readers may overlook these potentially important resources.

Thank you for this helpful comment. We will align the numbering of the supplementary materials accordingly and ensure that all supplementary figures and tables are properly referenced in the manuscript.

Technical comments:

- When citing, if applicable, it may be helpful to specify that exact page, figure or table for the reader's reference. For example, **L. 90**: (Table 5, Jagermeyr et al. 2015)

We will adjust that.

- **L. 156**: Please consider removing the comma following "*overuse*"

Will do.

- **L. 156**: It may improve readability to elaborate on the meaning of transgression: clarifying what it implies when a threshold has been transgressed.

Thank you for this comment. In the revised manuscript, we have clarified what transgressions of EFRs would imply.

- **L. 264-266**: The sentence "*While irrigation reduces GWS (below the threshold of 0.2) on 13% of the global agricultural area, it thereby leads to an increase in areas experiencing moderate BWS as well as high BWS by 6%, respectively.*" may be confusing, as "respectively" suggests a missing distinction between moderate and high BWS. Consider rephrasing for clarity, e.g., "…by 6% and 6%, respectively" or "…both by 6%"

Thank you for this valuable remark. We will adjust the text to make the distinction between moderate and high BWS clearer in the revised version of the manuscript.

**References**

Brouwer, C., Prins, K., Kay, M. and Heibloem, M.: Irrigation Water Management: Irrigation Methods. Training manual no. 5, 1988.

Fader, M., Rost, S., Müller, C., Bondeau, A. and Gerten, D.: Virtual Water Content of Temperate Cereals and Maize: Present and Potential Future Patterns, Journal of Hydrology, 384, 3–4, 218–31, https://doi.org/10.1016/j.jhydrol.2009.12.011, 2010.

Fischer, G., Nachtergaele, F., Prieler, S., Teixeira, E., Tóth, G., van Velthuizen, H., Verelst, L. and Wiberg, D.: Global Agro-ecological Zones (GAEZ v3.0), IIASA, Laxenburg, Austria and FAO, Rome, Italy, 2012.

Jägermeyr, J., Gerten, D., Heinke, J., Schaphoff, S., Kummu, M. and Lucht, W.: Water savings potentials of irrigation systems: global simulation of processes and linkages, Hydrology and Earth System Sciences, 19, 3073–3091, https://doi.org/10.5194/hess-19-3073-2015, 2015.

Sauer, T., Havlík, P., Schneider, U. A., Schmid, E., Kindermann, G. and Obersteiner, M.: Agriculture and Resource Availability in a Changing World: The Role of Irrigation, Water Resources Research, 46, 6, https://doi.org/10.1029/2009WR007729, 2010.